

**"Shallow Geophysical Techniques to Investigate the Groundwater Table
at the Giza Pyramids Area, Giza, Egypt."**
*Sharafeldin M. Sharafeldin[1], Khalid S. Essa[1], Mohammed A. S. Youssef[2], and Zein E. Diab[1]*
*[1]Geophysics Dept, Faculty of Science, Cairo University*
*[2]Nuclear Material Authority, P.O. Box 530, Maadi, Cairo*
*shokryam@yahoo.com*

**ABSTRACT**

Geophysical studies were performed along selected locations across the Pyramids Plateau to investigate the groundwater table and the near aquifer, which harmfully affected the existed monuments of the Giza Pyramids and Sphinx. Electrical Resistivity Imaging (ERI), Shallow Seismic Refraction (SSR) and Ground Penetrating Radar (GPR) techniques were carried out along selected profiles in the plateau. Ten ERI, twenty six SSR and nineteen GPR profiles were performed at the sites. The ERI survey shows that, the groundwater table is at elevations varying from 13 to 18 m above the sea level (asl) and low resistivity values near the surface at the Great Sphinx. ERI profiles, which were applied southeast of the Middle Pyramid (Khafre), show high resistivity values near the surface, and water table is located at elevations ranging from 22 to 40 m asl, while the ERI profiles conducted south of Menkaure, show almost high resistivity near the surface. The groundwater table is located at elevations ranging between 45 and 58m asl. The aquifer layer shows electrical resistivities ranging between 10 and 50 Ohm.m. The considerable high change in the groundwater table is due to the rapid increases of topography from the Great Sphinx towards the Small Pyramids (Menkaure), where this part looks-like a scarp. The SSR Survey is transmitted to know the different velocities and types of the layers, which can help in knowing the saturated layers in the area. The GPR Survey is performed to delineate the water table, which gives good matching with the ERI results.

*Keywords: Groundwater, Electrical Resistivity, Seismic refraction, GPR.*

**I. INDRDUCTION**

In recent years, the Egyptian Great Pyramids of Giza and Great Sphinx area are suffering from the rising of groundwater table, due to the rate of water leakage from the surrounded areas, due to the increase of expansion in urban developments of the new cities, population blocks, gardens and agricultural expansion, as well as the water distribution from the nearby Nile River Canals, such as El-Mansoria Canal. This problem promoted the need to monitor the groundwater table in the Pyramids area, thus by using geophysical techniques we can evaluate the effect of groundwater on the Pyramids and Sphinx area. Shallow geophysical investigation techniques for site characterization and near-surface targets, with the integration of the available geological data were used to develop a 3D model for the hydrologic system at the Giza Pyramids area. The analysis and interpretation of this model lead to characterize the groundwater table in the



pyramids area, to define the possible sources of water leakage, and to evaluate the effect of the
groundwater on the pyramids and Sphinx area.

Geophysical studies play an important and effective role in groundwater investigation.

During the last few years, near-surface geophysics, especially Geoelectric Resistivity, Shallow
Seismic Refraction and Ground Penetrating Radar techniques have been widely used in Egypt to
characterize the groundwater table and the subsurface rock masses. The present work
demonstrates the integration of electrical resistivity Imaging (ERI), shallow seismic refraction
(SSR), and Ground Penetrating Radar (GPR) techniques to delineate the groundwater table in the
Giza Pyramids area, (Fig.1). The Giza Pyramids Plateau is composed of a limestone cliff,
changes abruptly from the other side to a sandy desert plateau. The Ancient Egyptians called this
place Imentet, "The West" or Kherneter, "the Necropolis". The three Giza Pyramids named
(Khufu), Chephren (Khafre), and Menkaure are located along this limestone Plateau.
**II. GEOLOGY OF THE AREA**

The Pyramids Plateau is formed from massive limestones and dolomites (nummulitic

wacke-packestones) of the Middle Eocene Mokattam Formation, which dips with about 5-10° to
the SE direction. Steep escarpments border the plateau to the north and east directions (Fig. 2).
Southwards, the Mokattam Formation is overlained by less resistant sandy marls, marls and
weakly cemented limestones (argillaceous mud-wackestones) of the Upper Eocene Maadi
Formation. The top unit of the Maadi Formation comprises several meters of massive, partly
dolomitized limestones (pack-grainstones) of the so-called "Ain Musa Bed". The Maadi
Formation shows a more gentle escarpment toward the Mokattam Formation, to the north and to
the eastern Nile valley alluvium. The present escarpments represent a Pliocene shoreline and
documented the transgression of the Early Pliocene Sea from the Mediterranean up to the pre-
Nile valley ("Eonile", Said 1981, 1982), after the largely continental Oligocene and Miocene
times (Blankenborn 1921 and Said 1962, 1981 and 1990). A thin wedge of the Pliocene
sediments rests discordantly on the Maadi Formation, but only veneer remains distinct against
the Mokattam escarpment (Fig. 2). The inferred fault along the Mokattam Formation of the
Pyramids plateau reflects the fracturing of the limestone.
**III. GEOPHYSICAL INVESTIGATION TECHNIQUES**
**III.1 Electrical Resistivity Imaging (ERI) Surveying and data acquisition**

Two-dimensional electrical resistivity imaging (tomography) surveys are usually carried

out, using a large number of electrodes, 24 or more, connected to a multi-core cable (Griffiths
and Barker 1993). Syscal-Pro resistivity meter, IRIS Instruments, France production, was



deployed at the site of the Giza Pyramids plateau using, the Dipole-dipole electrode array configurations using 24 metal electrodes, with electrode spacing of 5 meter and 120 meter cable. The length of spread is 115 m for each profile and maximum investigation depth is 23.5 m. The automatic sequence were designed for the Dipole-dipole electrodes array configurations, using Electre-Pro program, version V.2.02.0 of IRIS instruments, then uploading this sequence from the PC to the Syscal-Pro resistivity meter, using the USB Dongle cable. In the field, select the uploaded automatic sequence and start on the acquisition.

In the study area of the Giza Pyramids, ten electrical resistivity imaging profiles were performed to characterize the resistivity values of the area, and hence to locate the groundwater table. Table 1 and figure 3 show the location of the electrical resistivity imaging profiles conducted at the study area. The profiles started from the Great Sphinx, through the Middle Pyramid (Khafre) and end at the Small Pyramid (Menkaure) from the southern part of the Giza Pyramids area.

**III.1.2 Data processing and interpretation**

The acquired electrical resistivity imaging profiles were processed and interpreted, using Prosys II program (version V.3.02.08) of IRIS Instruments and Rse2Dinv (Version 3.59) program of Geotomo software, Malaysia origin. Prosys II program is used to damp the data of the geoelectric resistivity imaging from the Syscal-Pro resistivity meter to the PC, using the USB Dongle cable, and utilized to filter and exterminate bad and noisy data acquired in the field. Res2Dinv program applies the least square inversions on the data exported from the Prosys II program, where the resistivity is plotted on a logarithmic scale function of the depth of the subsurface.

Ten electrical resistivity imaging (ERI) profiles were performed over the study area of the Giza Pyramids plateau. The topographic elevation is considered for each ERI profile and fed to the Res2Dinv program. The interpretations of the ERI1 to ERI3 profiles, which were taken beside the Great Sphinx, shows low resistivity values near the surface and shallow water table, which lies at elevations ranging from 13 to 18 m asl. The interpretation of the ERI4 and ERI5 profiles, which were located southeast of the Middle Pyramid (Khafre), shows high resistivity values near the surface, where the water table is followed at elevations range from 15 to 43 m (Figs. 4a and 4b). The analysis of the ERI8 to ERI10 profiles, which were conducted south of the Small Pyramid (Menkaure), shows almost high resistivity near the surface. The water table is located at elevations ranging between 45 and 58 m (Figs. 4c, 4d and 4e, respectively). These ERI Models reveals mostly four layers in average in most parts of the study area.





**III.2 Shallow Seismic Refraction (SSR)**

Compressional waves or (P-waves) are used almost exclusively in the seismic exploration for both seismic reflection and refraction. Especially at shallow depths, we primarily are concerned with P-wave velocities in the rocks (consolidated materials) and sediments (unconsolidated materials). In shallow refraction work, the P-wave velocities are often sufficient to describe the ground layers in terms of dry and wet overburden, and fresh and weathered bedrock. There are no unique velocity values for rocks or sediments; however, a few general rules are suggested by these values (Burger 1992). The water saturation, porosity, weathering, fracturing and compaction are factors affecting the layer velocities.

In the present study, however, the main target for applying the seismic refraction technique is to determine the seismic velocities and thicknesses of the different successive layers and to trace the lateral distribution of the subsurface layers throughout the investigated area. Depth-velocity models are constructed; these models reflect the number of layers penetrated by the seismic waves. Also, the type of lithology of each layer and water table within layer are determined, according to the values of velocities of the seismic waves through layers.

*III.2.1 Refraction data acquisition and survey parameters*

Twenty-six shallow seismic refraction profiles were acquired at the study area (Fig. 3). OYO McSEIS-SX seismograph, of 24 geophones-channels, was deployed at the studied site to collect the seismic refraction data. A sledge hammer (of 10 Kg) and an iron plate are used as P-wave seismic source. The used inter-receiver distance is 5 m. The numbers of shots are 5 shots per spread. Two off-set shots (each 20 meters from each end), forward (5 m from the first geophone), reverse (5 m from the last geophone) and a split spread shot. The spread, performed by seismograph covers 115 m. The nature of these important historical and touristic site in Egypt, where a huge number of visitors and human activities existed in the site, is imposing a considerable amount of noises to the recorded data. These noises were minimized as possible by using the internal frequency domain filter applied by the seismic seismograph. To enhance the data quality and decrease the random noises, each shot was repeated several times and stacked.

*III.2.2 Refraction data interpretation*

The first arrivals of the collected P-waves are picked, using Pickwin of SeisImager software version 4.2 (OYO 2011). These picked data are interpreted to get the depth-velocity models, by applying appropriate inversion techniques. Figures (5, 6, 7 and 8), respectively show examples of the interpreted seismic refraction profiles conducted at the site area. A three layers model assumed to represent the subsurface succession with the inverted velocities and thicknesses. The top most layer exhibits a velocity range of 400-900 m/s, and is correlated with



loose dry sand, fill and debris (which is corresponding to the first and second layer of electrical
resistivity model). The thickness of this layer ranges between 2 and 5 meters. The second layer
shows a velocity range between 1200 and 2400 m/s, this layer is correlated with wet and
saturated sand (which is corresponding to the third layer of electrical resistivity model). The
thickness of this layer varies from 10 to 20 meters. The third layer shows a higher domain of
velocity, where it ranges between 2800 and 3800 m/s, which can be correlated to marly
limestone and limestone (corresponding to the forth layer of electrical resistivity model), which
is considered as the aquifer layer, Table 1.
**III.3 Ground Penetrating Radar (GPR) techniques**
Ground-penetrating radar (GPR) is an effective tool to visualize the structure of the
shallow subsurface. Ground penetrating radar (GPR) has become a popular tool in the
environmental and engineering studies for the near-surface targets (Jol and Bristow 2003). It is a
non-invasive geophysical technique designed primarily for subsurface investigation (Neal 2004;
Comas et al. 2004). A GPR system detects changes in the electrical properties of the shallow
subsurface, using discrete pulses of high frequency electromagnetic (EM) energy, usually in the
10-1000 MHz range (Neal 2004).
The technique has been successfully applied in a wide range of environmental studies,
however an understanding of the capabilities and limitations of GPR is vital when considering
using the technique, with the quality of GPR results often being dependent on the surveyed
environment (Daniels 2004).
*III.3.1 GPR Surveying of the study area*
The GPR survey was carried out with MALA ProEx of Mala Geosciences, Sweden, using
100 MHz shielded antenna as a central frequency and data displayed, using a laptop computer.
Nineteen GPR profiles were performed along the study area of Giza Pyramids. The lengths of
GPR profiles range from 40 to 200 m, a total of about 2.5 kilometer of GPR surveys were
operated at the site. Locations and directions of the GPR profiles are viewed in Figure 3 and
Table-1. Surface topography elevations were taken into account in the GPR surveys, which can
be corrected in the radar processing, using static corrections. Wheel calibration was made near
the Great Sphinx along 30 m in distance, the velocity used in calibration is 100 m/μs and the No.
of stacking equal to 16. The depths of penetration vary from 8.5 to 20 m.
*III.3.2 GPR data Processing and analysis*
GPR data are subjected to a scheme of signal data processing, using Reflex-Win Version
6.0.9 software to enhance the quality of the gained data. The GPR data are displayed in cross
sections with the distance along the profile for the X-axis and with the two-way travel times of



the reflected GPR waves for the Y-axis. To convert the time sections into depth sections, an
average velocity of 0.1 m/ns was used, assuming a possible variation in depth of +/-10%., the
ground-vision of Mala Package and Reflex-W package are furnished to facilitate the processing
and interpretation of the acquired GPR data.
GPR data processing corrects the start time to compensate for air-wave and contact with
the ground, a DC-shift filter and an amplitude correction (Dewow) were applied to remove the
constant offset and compensate the loss spread and attenuation, respectively. Static corrections
were applied to the data to compensate for changes in the topographic elevations. A band-bass
filter, 2-D running average and gains function were applied to enhance the amplitudes of signals.
The background removal filter was applied to the data, the filter performs a subtraction of an
averaged trace. Deconvolution and stacking were performed to enhance the signal to noise ratio,
while Kirchhoff and diffraction migration processes were applied to the data to correct the
positions of reflection points. Muting also was introduced in some radar sections to remove the
bad data.
### III.3.3 Interpretation of GPR Data
The different colors of the radargram reflect the amplitudes of the reflected EM waves,
which are an indication of the change in the subsurface layers conductivities and dielectric
constants. GPR data resolve the locations of the layers boundaries as the dielectric constants of
the compositions changed, they are delineating the depths and extensions of the layers. GPR also
mapped the water table at the site. Nineteen GPR Profiles were conducted in the study area of the
Giza Pyramids (Fig. 3). In a way of knowing the groundwater table, the interpretation of GPR
profiles subdivided the area into four parts, according to the nearest one in distance and elevation
topography.
Area-I comprises the GPR Profile-1, which is located to the northwest of the Great
Pyramid. The site is a low land followed a scarp, it looks-like a wadi behind the plateau. The
interpreted GPR profile-1 reveals that, the water table uncertainly might be located at an
elevation of 20 to 21 meters asl, where the ground surface at 30 to 31 meters asl. Area-II
involves the GPR profile-2 to GPR profile-5 and the GPR profiles 9 and10. Such GPR profiles
are conducted along the low land of Nazlet El-saman village, beside the wall and east of the
Great Pyramid and Sphinx. The interpreted profiles, despite the noisy nature of the site due to the
walls and houses, reveals that the water table might be located at elevations ranging from 11.5
to17.5 meters asl. Area-III includes the GPR Profiles 6 to 8, which are located along the southern
eastern part of the study area, near El-Gabal El-Kebly. The water table is interpreted to locate at
elevation about 13 to 13.5 meters asl.  Area-IV comprises the GPR profiles 11 to 19, which are



located at the southern part of both the Middle Pyramid (Khafre) and the Smallest Pyramid
(Menkaure). The water table is located at elevations ranging from 14.5 to 68 meters asl. Table-1
summarizes the results of water table (WT) elevations to all the GPR profiles of the study area.
Figures (9a to 9d) view examples of the interpreted GPR profiles for each part.
*Table-1: summarized average ground elevations and the interpreted Groundwater table elevations with Coordinates for the different geophysical measurements*

| Geophysical survey | | | Coordinates | | Direction | Average Ground Elevation (m) | Average Water Table Elevation (m) |
|---|---|---|---|---|---|---|---|
| ERI No. | Seismic No. | GRP No. | X | Y | | | |
| | SSR1 | GPR1 | 320446 | 3317418 | NE-SW | 30 | 20 |
| | SSR2 | | 320356 | 3317386 | NE-SW | 31 | 21 |
| | SSR3 | GPR2 | 320228 | 3317226 | N-S | 26 | 17 |
| | SSR4 | | 320110 | 3317236 | N-S | 27 | 16 |
| | SSR5 | GPR3 | 320028 | 3317179 | NW-SE | 20 | 12.5 |
| | SSR6 | | 319845 | 3317010 | NW-SE | 20 | 11.5 |
| | SSR7 | GPR4 | 319739 | 3316914 | NW-SE | 21 | 13 |
| | SSR8 | GPR5 | 319582 | 3316803 | NW-SE | 18 | 11.5 |
| | SSR9 | GPR6 | 319392 | 3316898 | E-W | 17 | 13.5 |
| | SSR10 | GPR7 | 319238 | 3316958 | E-W | 17 | 13 |
| | SSR11 | | 319403 | 3318602 | E-W | 18 | 13.5 |
| | SSR12 | GPR8 | 319344 | 3318544 | E-W | 18 | 13 |
| ERI 1 | SSR13 | GPR9 | 320463 | 3317802 | N-S | 21 | 15-15.5 |
| ERI 2 | SSR14 | GPR10 | 320455 | 3317716 | NW-SE | 20 | 15-15.5 |
| ERI 3 | SSR15 | GPR11 | 320581 | 3317211 | NE-SW | 20 | 14-15 |
| ERI 4 | SSR16 | GPR12 | 320542 | 3317251 | NW-SE | 27 | 21-22 |
| ERI 5 | SSR17 | GPR13 | 320462 | 3317336 | NE-SW | 35 | 27-28 |
| ERI 6 | SSR18 | GPR14 | 320394 | 3317358 | N-S | 42 | 37 |
| ERI 7 | SSR19 | GPR15 | 320753 | 3316848 | NE-SW | 50 | 42-43 |
| ERI 8 | SSR20 | GPR16 | 320713 | 3316845 | N-S | 58 | 47-48 |
| ERI 9 | SSR21 | GPR17 | 320667 | 3316846 | NW-SE | 64 | 57 |
| ERI10 | SSR22 | GPR18 | 320625 | 3316845 | NW-SE | 66 | 59 |
| | SSR23 | GPR19 | 320441 | 3317414 | NW-SE | 75 | 68 |
| | SSR24 | | 320370 | 3317381 | NW-SE | 81 | 70 |
| | SSR25 | | 320284 | 3317316 | NE-SW | 99 | 90 |
| | SSR26 | | 320169 | 3317128 | NE-SW | 102 | 93 |

**IV. INTEGRATION OF THE DIFFERENT GEOPHYSICAL TECHNIQUES**
*IV.1 Comparison among the geophysical results at Area-I*
The correlation among the SSR and GPR data at Area-I of the study area of Giza
Pyramids plateau (Fig. 3) shows that, the results obtained from SSR and GPR data are were quite
matched, where the interpreted water table (WT) from the SSR is located at elevation of 21 m,
while the water table interpreted from the GPR is located at elevation of 20 m.

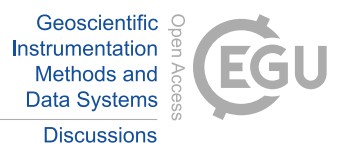



### IV.2 Comparison among the geophysical results at Area-II

The correlation between ERI, SSR and GPR data from some selected profiles at Area-II
of the study area of Giza Pyramids plateau (Fig. 3) views relatively the same results, where the
interpreted water table (WT) from the ERI is at elevation 15.5 m, while the water table
interpreted from the SSR is located at elevation 15 m, and the interpreted water table from the
GPR locates at elevation 15 m.

### IV.3 Comparison among the geophysical results at Area-III

     The correlation among the SSR and GPR data at Area-III of the study area of Giza
Pyramids plateau (Fig. 3), exhibits good matching among the different techniques, where the
interpreted water table (WT) from the SSR is at elevation 13.5 m, while the water table
interpreted from the GPR is located at elevation 13 m.

### IV.4 Comparison among the geophysical results at Area-IV

     The correlation among the ERI, SSR and GPR data from some selected profiles at Area-
IV of the study area of Giza Pyramids plateau (Fig. 3) gives results were quite matched, where
the interpreted water table (WT) from the ERI is at elevation 22 m, while the water table
interpreted from the SSR is located at elevation 22 m, and the interpreted water table from the
GPR locate at elevation 22 m. Figure 10 views an example of the correlations among the ERI,
SSR and GPR data from some selected profiles at Area-IV of the study area of Giza Pyramids
plateau.

### IV.5 2-D and 3-D presentation of the geophysical interpretation

     From all the geophysical techniques (ERI, SSR and GPR) that applied at the study area of
Giza Pyramids, the groundwater table elevations are interpreted and shown in Table-1 and Figure
11a. Figure 11a represents the groundwater elevations map from the geophysical surveys, which
were applied at the study area of Giza Pyramids in 2016, posting on it the groundwater levels
from the borehole Piezometers, which were installed by Cairo University in 2008 prior to the
activation of the dewatering system by AECOM 2010. All the previous geophysical surveys
conducted in 2016 show that, the pumping system by AECOM 2010 has good effect in lowering
the groundwater levels form the piezometer No.1 to the piezometer No.11, but from piezometer
No.12 to piezometer No.16, which represent the area concentrated around the Great Sphinx, still
have high groundwater levels and need more withdrawal to the groundwater in this part. Also we
recommended, making ceil or barrier in this part because of the surface is saturated with water,
which may be due to lake of the impervious clay in the subsurface in this region. Figure 11b
integrates the different geophysical techniques of the ERI, SSR and GPR into a 3D Model. The
3D model illustrates the topography, number of layers and the interpreted groundwater table
elevations above the sea level of the study area of the Giza Pyramids plateau.





**V. Conclusions**

The interpretation of the electrical resistivity imaging (ERI) survey, the aquifer layer shows electrical resistivities ranging between 10 and 50 Ohm.m. The imaging profiles near the Great Sphinx show the groundwater table at elevations varying from 13 to 18 m asl. The imaging profiles applied southeast of the Middle Pyramids (Khafre) show high resistivity values near the surface, and the groundwater table is located at elevations range from 22 to 40 m asl. The imaging profiles conducted to the south of the Small Pyramids (Menkaure) reveal almost high resistivity near the surface, where the groundwater table is located at elevations varying between 45 and 58 m asl.

Twenty six shallow seismic refraction (SSR) spreads were conducted across the study area of Giza Pyramids plateau, with spreads are ranging in length between 95 and 155 meters. A model of three layers assumed for the shallow section of the study area. The top most layer exhibits a velocity range of 400-900 m/s, this layer is correlated with loose dry sand and fill, with a thickness ranges between 2 and 6 meters. The second layer shows a velocity range between 1200 and 2800 m/s, this layer is correlated with wet and saturated sand, with a thickness varies from 5 to 12 meters, where the groundwater level is raised from the deep aquifer of the limestone up in the second layer. The third layer shows a wide variation in velocity, where it ranges between 2400 and 3950 m/s, which can be correlated with marly limestone and limestone.

Nineteen GPR profiles were performed along the study area of Giza Pyramids plateau, with a total length of about 2.5 kilometer of GPR surveys. In Area-I, the groundwater table is interpreted at elevations ranging between 20 and 21 m. In Area-II, the groundwater table is interpreted at elevations varying from 11.5 to 17.5 m. In Area-III, the groundwater table is interpreted at elevations ranging between 13 to 13.5 m. In Area-IV, the groundwater table is interpreted at elevations varying from 14.5 to 22.5 m around the Sphinx part and at the part of high topography near the Pyramids of Khafre and Menkaure, where the groundwater table is interpreted at elevations of 32 to 68 m asl.

The groundwater table is interpreted from the acquired geophysical data along the conducted profiles to be at the following elevations: Area-I: The groundwater table is located at elevations of 20 to 21 meter asl; Area-II: The groundwater table is located at elevations of 11.5 to 17.5 meters asl; Area-III: The groundwater table is located at elevations of 13 to 13.5 meters asl and Area-IV: The groundwater table is located at elevations of 14 to 58 meters asl.

It is evident that, the water table rises up the marly limestone layer of the Plateau, which is the formation of the Pyramids, that causes a serious problem, due to the corrosion and dissolution of the monuments. It is recommended to lower the water table to a level below the marly limestone to protect the existed monuments.

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
