# Peer review of ""Shallow Geophysical Techniques to Investigate the Groundwater Table at the Giza Pyramids Area, Giza, Egypt." Sharafeldin M. Sharafeldin1, Khalid S. Essa1, Mohammed A. S. Youssef2, and Zein E. Diab1"

_Geoscientific Instrumentation, Methods and Data Systems, 2017_

## Short Comment (SC1) · 24 Jan 2018

Dear authors, I have read your manuscript and I found it is interesting. I have some modifications: 1- where the figures of the manuscript, I did not find them. 2- the manuscript needs some moderate language revision.

---

## Author Comment (AC1) · 4 Feb 2018

Dear Sirs,

We would like to thank Prof. Morsy, the reviewer for his keen interest, valuable comments on the manuscript, and improvements to this work.

we have corrected, modified and inserted the missing figures on the manuscript. we have highlighted our changes by red color in the revised version.

we have uploaded the revised version as (Pdf file) using the Supplement button.

With my bets regards. Mohamed Shokry

---

## Author Comment (AC2) · 4 Feb 2018

PLEASE SEE ATTACHED FILE

Please also note the supplement to this comment:
https://www.geosci-instrum-method-data-syst-discuss.net/gi-2017-48/gi-2017-48-AC2-supplement.pdf
* * *
[Figure]

**Shallow Geophysical Techniques to Investigate the Groundwater Table
at the Giza Pyramids Area, Giza, Egypt**

*S. M. Sharafeldin[1,3], K. S. Essa[1], M. A. S. Youssef[2], H. Karsli[3], and Z. E. Diab[1], N. Sayil[3]*
[1]*Geophysics Dept, Faculty of Science, Cairo University*
[2]*Nuclear Material Authority, P.O. Box 530, Maadi, Cairo*
[3]*Geophysical Engineering Dept, KTU, Turkey*
*shokryam@yahoo.com*

**ABSTRACT**

The near surface groundwater aquifer that threatened the Great Giza Pyramids of Egypt, investigated using integrated geophysical surveys. Ten Electrical Resistivity Imaging, 26 Shallow Seismic Refraction and 19 Ground Penetrating Radar surveys conducted in the Giza Pyramids Plateau. Acquired data of each method subjected to state- of- the art processing and modeling techniques. A three-layer model depicts the subsurface layers and better delineates the groundwater aquifer and water table elevation. The aquifer layer resistivity ranges between 40-80 $\Omega$m and seismic velocity of 1500-1800 m/s. The average water table elevation is about +15 meters which is safe for Sphinx Statue, and still subjected to potential hazards from Nazlet Elsamman Suburban. Shallower water table in Valley Temple and Tomb of Queen Khentkawes detected to be between 14.5-15m represent a sever hazards. Perched ground water table detected in elevated topography to the west and southwest might be due to runoff and capillary seepage.

*Keywords: Groundwater, Electrical Resistivity, Seismic refraction, GPR.*

**I. INDRDUCTION**

In recent years, the 4500 years old Giza Great Pyramids (GGP) of Egypt; Cheops (Khufu), Chephren (Khafre), Menkaure and Sphinx statue; threatened from the rising groundwater table resulted from the water leakage of the suburban, irrigation canals and mass urbanization surrounding the GGP. This problem promoted the need to use non-destructive near surface geophysical techniques integrated with available borehole hydrogeological data to investigate and characterize the groundwater occurrences in the GGP. The GGP located in the southwestern part of the Greater Cairo Region (Fig. 1). Geologically, the Giza Pyramids Plateau composes mainly of white limestone, cream and yellow argillaceous limestone and dark grey dolomitic limestone of Middle-Upper Eocene age. The plateau rocks are commonly interbedded with thin marl layers in their upper part, which dips with about 5-10° to the SE direction. Steep escarpments border the plateau to the north and east directions as shown in Fig. 2 (Yehia, 1985; Mahmoud and Hamdan, 2002). Two regional groundwater aquifers underlie the sphinx (Fig. 3), the Quaternary aquifer of the Nile alluvium, consists of graded sand and gravel with

**Fig. 1.** final revised version include figures and replies

**Supplement:**

**Shallow Geophysical Techniques to Investigate the Groundwater Table at the Giza Pyramids Area, Giza, Egypt**

*S. M. Sharafeldin[1,3], K. S. Essa[1], M. A. S. Youssef[2], H. Karsli[3], and Z. E. Diab[1], N. Sayil[3]*
[1]*Geophysics Dept, Faculty of Science, Cairo University*
[2]*Nuclear Material Authority, P.O. Box 530, Maadi, Cairo*
[3]*Geophysical Engineering Dept, KTU, Turkey*
*shokryam@yahoo.com*

**ABSTRACT**

The near surface groundwater aquifer that threatened the Great Giza Pyramids of Egypt, investigated using integrated geophysical surveys. Ten Electrical Resistivity Imaging, 26 Shallow Seismic Refraction and 19 Ground Penetrating Radar surveys conducted in the Giza Pyramids Plateau. Acquired data of each method subjected to state- of- the art processing and modeling techniques. A three-layer model depicts the subsurface layers and better delineates the groundwater aquifer and water table elevation. The aquifer layer resistivity ranges between 40-80 Ωm and seismic velocity of 1500-1800 m/s. The average water table elevation is about +15 meters which is safe for Sphinx Statue, and still subjected to potential hazards from Nazlet Elsamman Suburban. Shallower water table in Valley Temple and Tomb of Queen Khentkawes detected to be between 14.5-15m represent a sever hazards. Perched ground water table detected in elevated topography to the west and southwest might be due to runoff and capillary seepage.

*Keywords: Groundwater, Electrical Resistivity, Seismic refraction, GPR.*

**I. INDRDUCTION**

In recent years, the 4500 years old Giza Great Pyramids (GGP) of Egypt; Cheops (Khufu), Chephren (Khafre), Menkaure and Sphinx statue; threatened from the rising groundwater table resulted from the water leakage of the suburban, irrigation canals and mass urbanization surrounding the GGP. This problem promoted the need to use non-destructive near surface geophysical techniques integrated with available borehole hydrogeological data to investigate and characterize the groundwater occurrences in the GGP. The GGP located in the southwestern part of the Greater Cairo Region (Fig. 1). Geologically, the Giza Pyramids Plateau composes mainly of white limestone, cream and yellow argillaceous limestone and dark grey dolomitic limestone of Middle-Upper Eocene age. The plateau rocks are commonly interbedded with thin marl layers in their upper part, which dips with about 5-10° to the SE direction. Steep escarpments border the plateau to the north and east directions as shown in Fig. 2 (Yehia, 1985; Mahmoud and Hamdan, 2002). Two regional groundwater aquifers underlie the sphinx (Fig. 3), the Quaternary aquifer of the Nile alluvium, consists of graded sand and gravel with intercalations of clay lenses at different depths exhibit water table at depth ranges between 1.5 to 4 meters bgs. The second aquifer is fissured carbonate aquifer that covers the area below the Pyramids Plateau and the Sphinx, where water table ranges in depth of 4 – 7 m bgs. The recharge of the aquifer below Sphinx area occurred mainly through water system leakage, Irrigation and massive urbanization (AECOM, 2010; and El-Arabi et al., 2013).

Many geophysical studies carried out in the GGP mostly for archaeological exploration and investigations (e.g., Dobecki, T. L., 2005; Abbas et al., 2009 and 2012). Geophysical studies have an effective contribution in characterizing groundwater aquifers especially geoelectrical resistivity, seismic refraction and ground penetrating radar techniques. Sharafeldin et al. (2017) studied the occurrence of the ground water table in GGP using combined VES, ERI and GPR to investigate the groundwater table in the area. The present work implemented an integration of Electrical Resistivity Imaging (ERI), Shallow Seismic Refraction (SSR), and Ground Penetrating Radar (GPR) techniques to depict the groundwater table and characterize the aquifer in the Giza Pyramids area. Figure-4 represents the locations of different surveys conducted in the GGP.

**II. Method**

**II.1 Electrical Resistivity Imaging (ERI) Surveys**

Two-dimensional electrical resistivity imaging (tomography) surveys are usually carried out, using a multi-electrode system, 24 or more, connected to a multi-core cable (Griffiths and King, 1965). Syscal-Pro resistivity meter, IRIS Instruments, France, was deployed at the site of the GGP using 24 multi-electrode dipole-dipole array configuration with 5m electrode spacing. The length of spread is 115m for each profile and attains 23.5 m maximum depth of investigation. Ten ERI profiles were performed to characterize the subsurface layers resistivities to delineate the groundwater aquifer (Fig. 4). The topographic elevation of each electrode is considered along ERI profile and fed to the Res2Dinv program. The acquired ERT data were processed using, Prosys II software of IRIS Instruments, to filter and exterminate bad and noisy data acquired in the field and produced the pseudo resistivity sections. The RES2DINV software implemented to invert collected data along conducted ERT profiles (Loke, and Barker, 1996; Loke, 2012). This software works based upon automatically subdividing the subsurface of desired profile into several rectangular prisms and then applies an iterative least-squares inversion algorithm for solving a non-linear set of equations to determine apparent resistivity values of the assumed prisms while decreasing the misfit values between the predicted and the measured data. Samples of interpreted data are shown in Figures 5 to 10.

**II.2 Shallow Seismic Refraction (SSR)**

Seismic refraction is widely used in determining the velocity and depth of weathering layer, static corrections for the deeper reflection data. It is also employed in civil engineering for the bedrock investigations and large scale building construction. It is also used in groundwater investigations, detection of fracture zones in hard rocks, examining stratigraphy and sedimentology, detecting geologic faults, evaluating karst conditions and for hazardous waste disposal delineation (Steeples, 2005; Stipe, 2015). A refraction technique is widely developed for characterizing the groundwater table (Grelle and Guadagno, 2009). Particularly, the unsaturated soil followed by saturated soil can be separated by a refracting interface or surface (Haeni, 1988). The seismic velocity values for the depth estimation of the groundwater can be used as an indicator for water saturation. The values of VP velocity are not uniquely correlated to the aquifer layer, but many authors related the P-wave velocities around 1500 m/s to represent a saturated layer (Grelle and Guadagno, 2009). The tomographic studies view that the water table corresponds to a P-wave velocity values of 1100 to1200 m/s (Azaria et al., 2003; Zelt et al., 2006).

Twenty-six SSR profiles were acquired at GGP (Fig. 4). 24 geophones-channels OYO McSEIS-SX seismograph was deployed in the GGP site to collect the seismic refraction data with geophone spacing is 5m. 10Kg sledge hammer and an iron plate are used to generate seismic P-wave. Five shots per spread were gathered, two off-set forward and reverse, and a split spread shot. The spread length covers 115m. Due to the historical and touristic nature of the site, a considerable amount of noise is imposing to the recorded data. These noises were minimized as possible by using the internal frequency domain filter and stacking of several shots during data acquisition. The first arrival times were picked using SeisImager software version 4.2 of OYO. The time-distance curve constructed and initial model for seismic tomography inversion of velocity and depth of the layered earth. Tomographic inversion; generate initial model from the velocity model obtained by the time-term inversion, then applying the inversion, which iteratively traces rays through the model with the goal of minimizing the RMS error between the observed and calculated travel-times curves (Schuster, 1998). SeisImager utilize a least squares approach for the inversion step (Zhang and Toksoz, 1998; Sheehan et al., 2005; Valenta, 2007). A three layers model assumed to represent the subsurface succession with the inverted velocities and thicknesses. The top most layer exhibits a velocity range of 400-900 m/s, and thickness of 2 and 5 meters, is correlated with loose dry sand, fill and debris. The second layer shows a velocity range between 1200 and 2400 m/s with 10 to 20 m thick. This layer is correlated with wet and saturated sand and fractured limestone. The third layer shows a higher domain of velocity, where it ranges between 2800 and 3800 m/s, which can be correlated to marly limestone and limestone.

Samples of interpreted data are shown in Figures 5 to 10.

**II.3 Ground Penetrating Radar (GPR) techniques**

GPR is a non-invasive geophysical technique and effective tool to visualize the near surface structure of the shallow subsurface and widely used to solve the environmental and engineering problems (Jol and Bristow, 2003; Comas et al., 2004; Neal, 2004). GPR is a site- specific technique that imposed a vital limitation of the quality and resolution of the acquired data (Daniels, 2004). The GPR surveys were carried out using 100 MHz shielded antenna of

MALA ProEx GPR. 19 GPR profiles were performed along selected locations in the study area (Figure 4). The lengths of GPR profiles range from 40 to 200 m according to the space availability with a total of total GPR survey of about 2.5 kilometer. Wheel calibration was made near the Great Sphinx along 30 m in distance, the velocity used in calibration is 100 m/μs using unshielded Puls Echo GPR. Harari (1996) showed that the groundwater table can be detected easily with a discerning selection of the antenna frequency and he observed that the lower frequency antenna (e.g.100 MHz) was more effective for locating the groundwater table depth.

Several basic processing techniques can be applied to GPR raw data stating from DC-shift to migration (Annan, 2005; Benedetto et al., 2017). All GPR sections along 19 profiles were processed to delineate subsurface layering and ground water elevation in the study area. To acquire better results, appropriate processing sequence of GPR data was applied to facilitate interpretation of radargram sections using REFLEXWIN V. 6.0.9 software. Time-zero correction filter first applied to all raw GPR data. Dewow Filter was applied to remove direct current and very low frequency components. A band-pass filter used to improve the visual quality of the

GPR data. Gain recovery applied to enhance the appearance of later arrivals because the effect of signal attenuation and geometrical spreading losses (Cassidy, 2009). Running average filters was the last filter applied. Samples of interpreted data are shown in Figures 5 to 10.

*III. Results and discussion*

The integrated interpretation of the SSR, ERI and GPR surveys supported a three layers model assumed to represent the subsurface succession with the inverted velocities, resistivities and thicknesses. The top most layer exhibits a velocity range of 400-900 m/s and a resistivity values varies between 10's to 100's Ohm.m and is correlated with heterogenous loose dry fill and debris of thickness ranges between 2 and 5 meters. The second layer shows a velocity range between 1200 and 2400 m/s and a resistivity values varies between 40 to 80 Ohm.m, this layer is correlated with wet and saturated sand and fractured limestone and the thickness varies between

10 to 15 meters. The third layer shows a high velocity ranges between 2800 to 3800 m/s and a resistivity values varies by changing the topographic elevation and marl intercalation in the limestone layer. GPR data delineated the subsurface succession and accurate detection of the water table in area near Sphinx, Valley Temple, Mastaba and Tombs. The ground water table detected ranges between 14-16 meters in these locations. As the ground relief increases toward the Mankaura Pyramids the water table is deeper and a perched water table detected in elevations between 22 to 45 meters.

Groundwater rise was detected in some locations of archaeological importance, these locations are Nazlet El-samman Village, Sphinx, Sphinx Temple, Valley Temple of Khafre, Central Field of Mastaba and Khafre Cause Way.

a- ***Nazlet El-samman Village*** is a suburban area located outside the core of the archeological site. The integration of different geophysical surveys conducted in this part, SSR-3 & 4, and GPR-2, revealed that the groundwater elevation at this part is about 16 m asl. The tomogram of SSR-3 & 4 show velocity of 1600-1800 m/s at elevation of 16 m asl. This elevation is fairly coinciding with the results of GPR-2 where a ground water level interpreted to be at 16 m elevation. The aquifer in this part is belonging to the Nile Alluvium Aquifer. The interpreted water table elevation between 16 and 17 m asl. This higher water table might affect the water table level below Sphinx area (Fig. 5).

b- ***Sphinx, Sphinx Temple, Valley Temple of Khafre, Central Field of Mastaba and Khafre Cause Way,*** this is the most important part of the study where the water appear on the surface at the Valley temple and surrounding area of the Sphinx. The locations of the surveys were chosen according to the limited space approved by the Pyramid Archaeological Authority. The locations of the conducted data are shown in (Fig.4). Survey shows good matching between the different techniques, where the correlation between different surveys results, revealed that groundwater elevation between 14-15 m asl. This level is lower than the suburban area of Nazlet El-samman, which might indicate a recharge of the aquifer below Sphinx and increase capillary water rise.

***Sphinx and Sphinx Temple,*** GPR-9, SSR-13 and ERI-1 conducted in front of Sphinx and Sphinx Temple. The integration of these surveys in front of Sphinx Temple, the groundwater elevation is about 14.5-15.5 m asl, as shown in Figure 6.

***Valley Temple of Khafre and central field of Mastaba,*** GPR profiles 3, 4, 5, 10 and 11; SSR profiles 5, 6, 7, 8 and 14; and ERI 2. The integration of this surveys in front of Valley Temple of Khafre and central field of Mastaba, the groundwater elevation is about 14-15 m asl as shown in Figure 7.

   ***Tomb of queen Khentkawes,*** GPR-11; SSR-15; and ERI-3 conducted near the Tomb.

   Figure 7 shows the surveys conduct near the site. The integration of this surveys in front

   of Valley Tomb of queen Khentkawes, the groundwater elevation is about 14.5-15 m asl.

   ***Valley Temple of Menkaure,*** GPR-12; SSR-16; and ERI-4 conducted near the Temple.

   The integration of these surveys in front of Valley Temple of Menkaure, the groundwater

   elevation is about 16.5-17 m asl. GPR profiles might detect the perched ground water

   table at shallower depth from ground level (Fig. 8).

   ***Cause way to Menkaure Pyramid***, show high resistivity value near the surface, and water

   table located at elevation ranges from 22 to 24 m asl. ***Menkaure Queens Pyramids and***

   ***Menkaure Quarry,*** where the surveys in this part conducted at higher topographic relief,

   the correlation of the different techniques revealed that the water table might be

   interpreted at elevations 45-58 m asl. This might detect the perched ground water table at

   shallower depth from ground level (Figs. 9 and 10).

Figure 11 represents a cross-section, using the ERT and GPR data, to illustrate the difference of groundwater table elevation between the Great Sphinx to the small pyramids of Menkaure that indicates the increase of groundwater elevation from west to east. As the average water table elevation to be about 15 m bsl, the water table to the west can be considered as perched water table to due leakage, surface runoff and capillary and fracture seepage. Figure 12 represents the compiled groundwater table elevation contour map from the geophysical surveys, overlay the groundwater table levels measured from some of the Piezometers installed by Cairo University (AECOM 2010). The present geophysical surveys proved that, the pumping system installed by

AECOM 2010 lowering the groundwater levels in some piezometer and a need of more pumping to compensate the recharge of the water leakage resulted from surrounding area of Sphinx.

Figure 13 shows a 3D representation of the groundwater system in Great Giza Pyramids Plateau and surrounding area.

**V. Conclusions**

The integrated interpretation of ERT, SSR and GPR surveys `conducted in Great Giza Pyramids site successfully investigate the groundwater aquifer and water table depth in Great Giza Pyramid and assist the hazards mitigation in a great historical heritage. An interpreted model consists of three layers assumed to depict the subsurface layers and better delineation of the aquifer layer associated with resistivity range of 40-80 $\Omega$m and seismic velocity of 1500-1800 m/s. The average water table depth is about 15m asl, which is safe for the Sphinx status where the foot at elevation of 20 m asl. The water table elevation increases in Nazlet Elsamman Village to 16m and causes leakage towards the Sphinx and Valley Temple which considered a serious hazard to the site. Tomb of Queen Khentkawes threatened by water leakage resulted from vegetation in old cemetery and nearby football field. A parched groundwater table might exist in elevated area toward west and southwest. A great care should be taken to the effect of massive urbanization to the west of the Great Giza Pyramids which might affect the groundwater model of the area. The dewatering system should be accomplished to avoid such hazards.

**Acknowledgements**

Authors would like to thank Prof. Jothiram Vivekanandan, Chief-Executive Editor, Prof. Andrea Benedetto, the Associate Editor and the reviewer for their constructive comments for improving our manuscript.

**References**

Abbas, A. M., Atya, M., EL-Emam, A., Ghazala, H., Shabaan, F., Odah, H., El-Kheder, I., and Lethy, A.: Integrated Geophysical Studies to Image the Remains of Amenemeht- II Pyramid's Complex in Dahshour Necropolis, Giza, Egypt. NRIAG, 2009. https://www.researchgate.net/publication/234180809.

Abbas, A. M., El-sayed, E. A., Shaaban, F. A., and Abdel-Hafez, T.: Uncovering the Pyramids-Giza Plateau in a Search for Archaeological Relics by Utilizing Ground Penetrating Radar. Journal of American Science, 8(2), 1-16, 2012.

AECOM, ECG, and EDG: Pyramids Plateau Groundwater Lowering Activity. Groundwater Modeling and Alternatives Evaluation. USAID Contract No EDH-I-00-08-00024-00-Order No.02, 2010.

Annan, A. P., [2005] Ground-penetrating radar. In Near surface geophysics, Butler DK (ed). Society of exploration geophysicists: Tulsa, Investigations in Geophysics 13, 357-438.

Azaria, A., Zelt, C. A., and Levander, A.: High-resolution seismic mapping at a groundwater contamination site: 3-D traveltime tomography of refraction data. EGS–AGU–EUG joint Assembly, Abstracts from the meeting held in Nice, 2003.

Benedetto, A., Tosti, F., Ciampoli, L. B., and D'Amico, F.: An overview of ground-penetrating radar signal processing techniques for road inspections. Signal Processing, 132, 201-209, 2017.

Cassidy, N. J.: Ground penetrating radar data processing, modelling and analysis. In Ground penetrating radar: theory and applications, Jol HM (ed). Elsevier:Amsterdam, 141-176, 2009.

Comas X., Slater L. and Reeve A.: Geophysical evidence for peat basin morphology and stratigraphic controls on vegetation observed in a northern peat land. Journal of Hydrology, 295, 173-184, 2004.

Daniels, D.J.: Ground penetrating radar (2nd edition). The Institution of Electrical Engineers: London, 2004.

Dobecki, T. L.: Geophysical Exploration at the Giza Plateau, Egypt aTen-Year Odyssey. Environmental & Engineering Geophysical Society (EEGS). 18th EEGS Symposium on the Application of Geophysics to Engineering and Environmental Problems, 2005.

El-Arabi, N., Fekri, A., Zaghloul, E. A., Elbeih, S. F., and laake A.: Assessment of Groundwater Movement at Giza Pyramids Plateau Using GIS Techniques. Journal of Applied Sciences Research, 9(8), 4711-4722, 2013.

Grelle, G. and Guadagno, F. M.: Seismic refraction methodology for groundwater level determination: "Water seismic index". Journal of Applied Geophysics 68, 301–320, 2009.

Griffiths D. H. and King R. F.: Applied geophysics for Engineering and geologists, Pergamon press, Oxford, New York, Toronto, 221p, 1965.

Harari, Z.: Ground-penetrating radar (GPR) for imaging stratigraphic features and groundwater in sand dunes. J. Appl. Geophys., 36, 43–52, 1996.

Jol, H. M. and Bristow C. S.: GPR in sediments: advice on data collection, basic processing and interpretation, a good practice guide. In Ground penetrating radar insediments, Bristow CS and Jol HM (eds). Geological Society: London, Special Publication 211; 9- 28, 2003.

Loke, M. H., and Barker, R. D.: Rapid least-squares inversion of apparent resistivity pseudo-sections by a quasi- Newton method. Geophysical Prospecting, 44 (1), 131–152, 1996.

Loke M. H.: Tutorial: 2-D and 3-D electrical imaging surveys. Course Notes, 2012.

Mahmoud, A. A., and Hamdan, M. A.: On the stratigraphy and lithofacies of the pleistocene sediments at Giza pyramidal area, Cairo, Egypt. Sedimentology of Egypt, 10, 145-158, 2002.

Neal A.: Ground-penetrating radar and its use in sedimentology: principles, problems and progress. Earth science reviews, 66, 261-330, 2004.

Schuster, G. T.: Basics of Exploration Seismology and Tomography. Basics of Traveltime Tomography. Stanford Mathematical Geophysics Summer School Lectures. 1998. (*http://utam.geophys.utah.edu/stanford/node25.html*).

Sharafeldin, M., Essa, K.S. , Sayıl, N. , Youssef, ., Diab, Z. E., and Karslı, H.: Geophysical Investigation Of Ground Water Hazards In Giza Pyramids And Sphinx Using Electrical Resistivity Tomography And Ground Penetrating Radar: A Case Study. Extended Abstract, 9th Congress of the Balkan Geophysical Society, Antalya, Turkey, DOI: 10.3997/2214-4609.201702549, 2017.

Sheehan, J. R., Doll, W. E., and Mandell, W. A.: An Evaluation of Methods and Available Software for Seismic Refraction Tomography Analysis. JEEG, 10 (1), 21–34, 2005.

Steeples, D. W.: Shallow Seismic Methods. In Y. Rubin, & S. S. Hubbard, Hydrogeophysics (pp: 215-251). Netherlands: Springer, 2005.

Stipe, T.: A Hydrogeophysical Investigation of Logan, MT Using Electrical Techniques and Sseismic Refraction Tomography. Degree of Master of Science in Geoscience: Geophysical Engineering Option. Montana Tech., 2015.

Valenta, J., and Dohnal, J.: 3D seismic travel time surveying – a comparison of the time- term method and tomography (an example from an archaeological site). Journal of Applied Geophysics, 63, 46-58, 2007.

Yehia A.: Geological structures of the Giza pyramids plateau. Middle East Res. Center, Ain Shams Univ., Egypt, Sci. Res. Series, 5, 100-120, 1985.

Zelt, A. C., Azaria, A., and Levander, A.: 3D seismic refraction travel time tomography at a groundwater contamination site. Geophysics, 58(9), 1314–1323, 2006.

Zhang, J., and Toksoz, M.: Nonlinear refraction traveltime tomography. Geophysics, 63(5), 1726–1737, 1998.

[Figure]

Fig. 1: Location map of the study area of Pyramids Plateau.

Fig. 2: Geologic map of the Giza Pyramid Plateau, Egypt. (Modified after Yehia, 1985).

[Figure]

Fig. 3 Ground water aquifers affected the Giza Pyramids Plateau (El-Arabi et al., 2013).

[Figure]

**Fig. 4: locations for the profiles and techniques used along the different parts of the Giza Pyramids plateau.**

[Figure]

**A-**

**B-**

**Fig. 5. SSR and GPR profiles in Nazlet El-semman**

**A- ERT1**

Line1.bin

Model resistivity with topography
Iteration 6 Abs. error = 5.0

Saturated Sediments

Water Table

Low high

16.3  28.8  50.8  89.7  158  279  493  871
Resistivity in ohm.m

**B-SSR13**

Scale = 1 / 1000

Velocity

(m/s)

Distance (m)

**C-GPR9**

Water table

**Fig. 6. ERI, SSR and GPR profiles in Sphinx and sphinx Temple**

**A- ERT2**

**B-SSR14**

**C-GPR5**

*Fig. 7. SSR and GPR profiles in Valley Temple of Khafre and central field of Mastaba*

**A- ERT3**

**B-SSR15**

**C-GPR11**

*Fig. 8. ERT, SSR and GPR profiles in Tomb of queen Khentkawes*

[Figure]

**Fig. 9. ERT, SSR and GPR profiles in Valley Temple of Menkaure.**

[Figure]

**Fig. 10. ERT, SSR and GPR profiles in _Cause way to Menkaure Pyramid._**

[Figure]

*Figure 11 Cross-section using the ERT data shows how the groundwater elevation change from Sphinx to Menkaure Pyramid.*

*Fig. 12: Groundwater elevations map from the ERI, SSR and GPR data taken across the study area of the Giza Pyramids plateau, including the installed piezometers and their groundwater levels by Cairo University 2008.*

[Figure]

*Fig. 13: 3D model of the Giza Pyramids Plateau, illustrating the groundwater table.*

**Author's response to the Associate Editor comment on the paper entitled "Shallow Geophysical Techniques to Investigate the Groundwater Table at the Giza Pyramids Area, Giza, Egypt" gi-2017-39**

**Authors: S. M. Sharafeldin, K. S. Essa, M. A. S. Youssef, H. Karsli, and Z. E. Diab, N. Sayil**

We would like to thank Prof. Lev Eppelbaum, Associate Editor, for his keen interest, valuable comments on the manuscript, and improvements to this work.

**Replies to the comments of the reviewer**

**Comment #1:-**
**"Dear authors, I have read your manuscript and I found it is interesting. I have some modifications:**

1- **where the figures of the manuscript, I did not find them.**

2- **the manuscript needs some moderate language revision".**

**Reply:**

Thank you. We have corrected, modified and added the missing figures.

**Thank you**

---

## Short Comment (SC2) · 5 Feb 2018

the authors improved the manuscript in a professional way. No, the manuscript is accepted as it is.

---

## Short Comment (SC3) · 5 Feb 2018

the authors improved the manuscript in a professional way. Now, the manuscript is accepted as it is.
* * *

---

## Referee Comment (RC1) · Anonymous Referee #3 · 14 Jun 2018

Authors present a case study dealing with a multi sensor approach in the assessment of the water table level in the Giza Plateau. The field data were collected by using 3 different geophysical techniques: ERI, SSR, GPR. Field setups and measurements procedures are quite well described; I suggest the authors to introduce additional information about the gauges calibration. The data processing and analysis is performed through existing software. It is not clear in the text the use of the boreholes data. The paper does not present novel tools or analysis techniques; furthermore the integration of data, collected through different instruments, is quite common. Despite this, the study can be interesting for the specific investigation site and for a cost-effective planning of future measurement campaigns. A more interesting data presentation could

be obtained by introducing the uncertainty in the analysis. The text is generally well written, but sometimes it is redundant. As noticed by the SC1, figures are not in the pdf.

---

## Author Comment (AC4) · 4 Jul 2018

Dear Sirs,

We would like to to thank Prof. Jothiram Vivekanandan, Chief-Executive Editor, Prof. Andrea Benedetto, the Associate Editor, and the reviewer for their constructive comments for improving our manuscript.

we have corrected, modified and inserted the missing figures on the manuscript. We have highlighted our changes by red color in the revised version.

We have uploaded the revised version as (Pdf file)including the authors response to the

reviewer comments using the Supplement button. Please Upload the newest version in your web site because the old version is in your system.

With my bets regards. Mohamed Shokry

Please also note the supplement to this comment:
https://www.geosci-instrum-method-data-syst-discuss.net/gi-2017-48/gi-2017-48-AC4-supplement.pdf

**Shallow Geophysical Techniques to Investigate the Groundwater Table at the Giza Pyramids Area, Giza, Egypt**

*S. M. Sharafeldin[1,3], K. S. Essa[1], M. A. S. Youssef[2*], H. Karsli[3], Z. E. Diab[1], and N. Sayil[3]*

*[1]Geophysics Department, Faculty of Science, Cairo University, Giza, P.O.12613, Egypt*
*[2]Nuclear Materials Authority, P.O. Box 530, Maadi, Cairo, Egypt*
*[3]Geophysical Engineering Department, KTU, Turkey*
*[*]shokryam@yahoo.com*

**ABSTRACT**

The near surface groundwater aquifer that threatened the Great Giza Pyramids of Egypt, was investigated using integrated geophysical surveys. Ten Electrical Resistivity Imaging, 26 Shallow Seismic Refraction and 19 Ground Penetrating Radar surveys were conducted in the Giza Pyramids Plateau. Collected data of each method evaluated by the state- of- the art processing and modeling techniques. A three-layer model depicts the subsurface layers and better delineates the groundwater aquifer and water table elevation. The aquifer layer resistivity and seismic velocity vary between 40-80 Ωm and 1500-1800 m/s. The average water table elevation is about +15 meters which is safe for Sphinx Statue, and still subjected to potential hazards from Nazlet Elsamman Suburban where a water table elevation attains 17 m. Shallower water table in Valley Temple and Tomb of Queen Khentkawes of low topographic relief represent a sever hazards. It can be concluded that perched ground water table detected in

**Shallow Geophysical Techniques to Investigate the Groundwater Table at the Giza Pyramids Area, Giza, Egypt**

*S. M. Sharafeldin[1,3], K. S. Essa[1], M. A. S. Youssef[2*], H. Karsli[3], Z. E. Diab[1], and N. Sayil[3]*
*[1]Geophysics Department, Faculty of Science, Cairo University, Giza, P.O.12613, Egypt*
*[2]Nuclear Materials Authority, P.O. Box 530, Maadi, Cairo, Egypt*
*[3]Geophysical Engineering Department, KTU, Turkey*
*[*]shokryam@yahoo.com*

**ABSTRACT**

The near surface groundwater aquifer that threatened the Great Giza Pyramids of Egypt, was investigated using integrated geophysical surveys. Ten Electrical Resistivity Imaging, 26

Shallow Seismic Refraction and 19 Ground Penetrating Radar surveys were conducted in the

Giza Pyramids Plateau. Collected data of each method evaluated by the state- of- the art processing and modeling techniques. A three-layer model depicts the subsurface layers and better delineates the groundwater aquifer and water table elevation. The aquifer layer resistivity and seismic velocity vary between 40-80 $\Omega$m and 1500-1800 m/s. The average water table elevation is about +15 meters which is safe for Sphinx Statue, and still subjected to potential hazards from Nazlet Elsamman Suburban where a water table elevation attains 17 m. Shallower water table in Valley Temple and Tomb of Queen Khentkawes of low topographic relief represent a sever hazards. It can be concluded that perched ground water table detected in elevated topography to the west and southwest might be due to runoff and capillary seepage.

**Shallow Geophysical Techniques to Investigate the Groundwater Table at the Giza**
**Pyramids Area, Giza, Egypt**
*S. M. Sharafeldin[1,3], K. S. Essa[1], M. A. S. Youssef[2]\*, H. Karsli[3], Z. E. Diab[1], and N. Sayil[3]*
*[1]Geophysics Department, Faculty of Science, Cairo University, Giza, P.O.12613, Egypt*
*[2]Nuclear Materials Authority, P.O. Box 530, Maadi, Cairo, Egypt*
*[3]Geophysical Engineering Department, KTU, Turkey*
*\*shokryam@yahoo.com*
**ABSTRACT**

The near surface groundwater aquifer that threatened the Great Giza Pyramids of Egypt, was investigated using integrated geophysical surveys. Ten Electrical Resistivity Imaging, 26

Shallow Seismic Refraction and 19 Ground Penetrating Radar surveys were conducted in the

Giza Pyramids Plateau. Collected data of each method evaluated by the state- of- the art processing and modeling techniques. A three-layer model depicts the subsurface layers and better delineates the groundwater aquifer and water table elevation. The aquifer layer resistivity and seismic velocity vary between 40-80 $\Omega$m and 1500-1800 m/s. The average water table elevation is about +15 meters which is safe for Sphinx Statue, and still subjected to potential hazards from Nazlet Elsamman Suburban where a water table elevation attains 17 m. Shallower water table in Valley Temple and Tomb of Queen Khentkawes of low topographic relief represent a sever hazards. It can be concluded that perched ground water table detected in elevated topography to the west and southwest might be due to runoff and capillary seepage.

**Supplement:**

7

**Shallow Geophysical Techniques to Investigate the Groundwater Table at the Giza Pyramids Area, Giza, Egypt**

S. M. Sharafeldin1,3, K. S. Essa1, M. A. S. Youssef2\*, H. Karsli3, Z. E. Diab1, and N. Sayil3

1Geophysics Department, Faculty of Science, Cairo University, Giza, P.O.12613, Egypt

2Nuclear Materials Authority, P.O. Box 530, Maadi, Cairo, Egypt

3Geophysical Engineering Department, KTU, Turkey

\*shokryam@yahoo.com

**9 ABSTRACT**

The near surface groundwater aquifer that threatened the Great Giza Pyramids of Egypt, 10 was investigated using integrated geophysical surveys. Ten Electrical Resistivity Imaging, 26 11 Shallow Seismic Refraction and 19 Ground Penetrating Radar surveys were conducted in the 12 Giza Pyramids Plateau. Collected data of each method evaluated by the state- of- the art 13 processing and modeling techniques. A three-layer model depicts the subsurface layers and 14 better delineates the groundwater aquifer and water table elevation. The aquifer layer resistivity 15 and seismic velocity vary between 40-80  $\Omega$ m and 1500-1800 m/s. The average water table 16 elevation is about +15 meters which is safe for Sphinx Statue, and still subjected to potential 17 hazards from Nazlet Elsamman Suburban where a water table elevation attains 17 m. Shallower 18 water table in Valley Temple and Tomb of Queen Khentkawes of low topographic relief 19 represent a sever hazards. It can be concluded that perched ground water table detected in 20 21 elevated topography to the west and southwest might be due to runoff and capillary seepage.

Keywords: Giza Pyramids, Groundwater, Electrical Resistivity, Seismic refraction, GPR.

**25 I. INDRDUCTION**

In recent years, the 4500 years old Giza Great Pyramids (GGP) of Egypt; Cheops 26 27 (Khufu), Chephren (Khafre), Menkaure and Sphinx statue; threatened from the rising groundwater table resulted from the water leakage of the suburban, irrigation canals and mass 28 29 urbanization surrounding the GGP. This problem promoted the need to use non-destructive near surface geophysical techniques integrated with available borehole hydrogeological data to 30 investigate and characterize the groundwater occurrences in the GGP. The GGP located in the 31 southwestern part of the Greater Cairo Region (Fig. 1). Geologically, the Giza Pyramids Plateau 32 33 composes mainly of white limestone, cream and yellow argillaceous limestone and dark grey dolomitic limestone of Middle-Upper Eocene age. The plateau rocks are commonly interbedded 34 with thin marl layers in their upper part, which dips with about 5-10° to the Southeast (SE) 35 direction. Steep escarpments border the plateau to the north and east directions as shown in Fig. 36 2 (Yehia, 1985; Mahmoud and Hamdan, 2002). Two regional groundwater aquifers underlie the 37

Sphinx (Fig. 3), the Quaternary aquifer of the Nile alluvium, consists of graded sand and gravel with intercalations of clay lenses at different depths exhibit water table at depth ranges between 1.5 to 4 meters below ground surface (bgs). The second aquifer is fissured carbonate aquifer that covers the area below the Pyramids Plateau and the Sphinx, where water table ranges in depth of 4 - 7 m bgs. The recharge of the aquifer below Sphinx area occurred mainly through water system leakage, Irrigation and massive urbanization (AECOM, 2010; and El-Arabi et al., 2013).

Many geophysical studies carried out in the GGP mostly for archaeological exploration 44 and investigations (e.g., Dobecki, T. L., 2005; Abbas et al., 2009 and 2012). Geophysical studies 45 have an effective contribution in characterizing groundwater aquifers especially geoelectrical 46 resistivity, seismic refraction and ground penetrating radar techniques. Sharafeldin et al. (2017) 47 48 studied the occurrence of the ground water table in GGP using combined VES, ERI and GPR to investigate the groundwater table in the area. The present work implemented an integration of 49 Electrical Resistivity Imaging (ERI), Shallow Seismic Refraction (SSR), and Ground Penetrating 50 Radar (GPR) techniques to depict the groundwater table and characterize the aquifer in the Giza 51 Pyramids area. The locations of different surveys conducted in the GGP are illustrated in Fig. 4. 52

**54 II. Method**

**55 II.1 Electrical Resistivity Imaging (ERI) Surveys**

Two-dimensional electrical resistivity imaging (tomography) surveys are usually carried out, using a multi-electrode system with, 24 electrodes or more, connected to a multi-core cable 57 (Griffiths and King, 1965). Syscal-Pro resistivity meter, IRIS Instruments, France, was deployed 58 59 at the site of the GGP using 24 multi-electrode dipole-dipole array configuration with 5m electrode spacing. The length of spread is 115m for each profile and attains 23.5 m maximum 60 depth of investigation. Ten ERI profiles were performed to characterize the resistivity of 61 subsurface layers to delineate the groundwater aquifer (Fig. 4). The topographic elevation of 62 each electrode is considered along ERI profile and linked to the Res2Dinv program. The 63 acquired ERT data were processed using, Prosys II software of IRIS Instruments, to filter and 64 exterminate bad and noisy data acquired in the field and produced the pseudo resistivity sections. 65 The Res2Dinv software implemented to invert the collected data along conducted ERT profiles 66 67 (Loke, and Barker, 1996; Loke, 2012). This software works based upon automatically subdividing the subsurface of desired profile into several rectangular prisms and then applies an 68 69 iterative least-squares inversion algorithm for solving a non-linear set of equations to determine apparent resistivity values of the assumed prisms while decreasing the misfit values between the 70 predicted and the measured data. Samples of interpreted data are shown in Figures 5 to 10. 71

**73**

**II.2 Shallow Seismic Refraction (SSR)**

Seismic refraction is widely used in determining the velocity and depth of weathering 74 layer, static corrections for the deeper reflection data. It is also employed in civil engineering for 75 the bedrock investigations and large scale building construction. It is also used in groundwater 76 investigations, detection of fracture zones in hard rocks, examining stratigraphy and 77 sedimentology, detecting geologic faults, evaluating karst conditions and for hazardous waste 78 disposal delineation (Steeples, 2005; Stipe, 2015). A refraction technique is widely developed 79 for characterizing the groundwater table (Grelle and Guadagno, 2009). Particularly, the 80 unsaturated soil followed by saturated soil can be separated by a refracting interface (Haeni, 81 82 1988). The seismic velocity values for the depth estimation of the groundwater can be used as an indicator for water saturation. The values of P-wave velocity are not uniquely correlated to the 83 aquifer layer, but many authors related the P-wave velocities around 1500 m/s to represent a 84 saturated layer (Grelle and Guadagno, 2009). The tomographic studies view that the water table 85 corresponds to a P-wave velocity values of 1100 to1200 m/s (Azaria et al., 2003; Zelt et al., 86 87 2006).

Twenty-six SSR profiles were acquired at GGP (Fig. 4). OYO McSEIS-SX seismograph 88 with 24 geophones and channels, was deployed in the GGP site to collect the seismic refraction 89 data with geophone spacing of 5m. Sledge hammer with 10Kg and an iron-steel plate are used to 90 91 generate seismic P-wave. Five shots per spread were gathered, two off-set forward and reverse, and a split spread shot. The spread length covers 115m. Due to the historical and touristic nature 92 93 of the site, a considerable amount of noise is imposing to the recorded data. These noises were minimized as possible by using the internal frequency domain filter and vertical stacking of 94 95 several shots during data acquisition. The first arrival times were picked using SeisImager software version 4.2 of Geometrics. Tomographic inversion; generate initial model from the 96 velocity model obtained by the time-term inversion, then applying the inversion, which 97 iteratively traces rays through the model with the goal of minimizing the RMS error between the 98 observed and calculated travel-times curves (Schuster, 1998). SeisImager utilize a least squares 99 approach for the inversion step (Zhang and Toksoz, 1998; Sheehan et al., 2005; Valenta, 2007). 100 A three layers model assumed to represent the subsurface succession with the inverted velocities 101 and thicknesses. The top most layer exhibits a velocity range of 400-900 m/s, and thickness of 2 102 and 5 meters, is correlated with loose dry sand, fill and debris. The second layer shows a velocity 103 range between 1200 and 2400 m/s with 10 to 20 m thick. This layer is correlated with wet and 104 saturated sand and fractured limestone. The third layer shows a higher domain of velocity, where 105

it ranges between 2800 and 3800 m/s, which can be correlated to marly limestone and limestone.

The calculated arrival time for the resulted model is compared with the measured arrival time
and RMS error is calculated and illustrated on modeled seismic profiles. Samples of interpreted data are shown in Figures 5 to 10.

II.3 Ground Penetrating Radar (GPR) techniques

GPR is a non-invasive and effective geophysical technique to visualize the near surface 111 structure of the shallow subsurface and widely used to solve the environmental and engineering 112 problems (Jol and Bristow, 2003; Comas et al., 2004; Neal, 2004). GPR is a site-specific 113 technique that imposed a vital limitation of the quality and resolution of the acquired data 114 (Daniels, 2004). The GPR surveys were carried out using 100 MHz shielded antenna of MALA 115 ProEx GPR. A total of 19 GPR profiles were performed along selected locations in the study 116 area (Figure 4). The GPR profiles range in lengths from 40 to 200 m, according to the space 117 availability, with a total GPR surveys of about 2.5 kilometer. Wheel calibration was carried out 118 near the Great Sphinx along 30 m in distance, the velocity used in calibration is 100 m/us 119 120 resulted from WAAR test using 100 MHz unshielded antenna of Puls-Echo GPR. Harari (1996) showed that the groundwater table can be detected easily with a discerning selection of the 121 antenna frequency and he observed that the lower frequency antenna (e.g.100 MHz) was more 122 effective for locating the groundwater table depth. Several basic processing techniques can be 123 applied to GPR raw data stating from DC-shift to migration (Annan, 2005; Benedetto et al., 124 2017). All 19 GPR profiles were processed to delineate subsurface layering and ground water 125 elevation in the study area. Appropriate processing sequence for GPR data was applied to 126 facilitate interpretation of radargram sections using REFLEXWIN V. 6.0.9 software (Sandmeier, 127 2012). Firstly time-zero correction, and then dewow filters to remove DC component and very 128 129 low frequency components were applied to all GPR data. Then, a band-pass filter was used to improve the visual quality of the GPR data, gain recovery was applied to enhance the appearance 130 131 of later arrivals because the effect of signal attenuation and geometrical spreading losses (Cassidy, 2009). Running average filters was the last filter applied. Some sections of interpreted 132 133 data are shown in Figures 5 to 10.

III. Results and discussion

The integrated interpretation of the SSR, ERI and GPR surveys supports a three layers model assumed to represent the subsurface succession with the inverted velocities, resistivities and thicknesses. The top most layer exhibits a velocity range of 400-900 m/s and a resistivity values varies between 10's to 100's Ohm.m and is correlated with heterogeneous loose dry fill and debris of thickness ranges between 2 and 5 meters. The second layer shows a velocity range 140 between 1200 and 2400 m/s and a resistivity values varies between 40 to 80 Ohm.m, this layer is correlated with wet and saturated sand and fractured limestone and the thickness varies between 141 10 to 15 meters. The third layer shows a high velocity ranges between 2800 to 3800 m/s and a 142 resistivity values varies by changing the topographic elevation and marl intercalation in the 143 limestone layer. GPR data delineated the subsurface succession and accurate detection of the 144 water table in area near Sphinx, Valley Temple, Mastaba and Tombs. The interpreted ground 145 water table elevation ranges between 14-16 meters in these locations. As the ground relief 146 increases toward the Mankaura Pyramids the water table is deeper and a perched water table 147 detected in elevations between 22 to 45 meters. 148

Groundwater rise was detected in some locations which have an archaeological
importance, these locations are Nazlet El-samman Village, Sphinx, Sphinx Temple, Valley
Temple of Khafre, Central Field of Mastaba and Khafre Cause Way.

- *a- Nazlet El-samman Village* is a suburban area located outside the core of the archeological site. The geophysical surveys SSR-3 & 4 and GPR-2 conducted in the area show a velocities of 1600-1800 m/s and interpreted water table at elevation of 16-17 m. This elevation is fairly matched with a nearest piezometers-6 and 7 in the area where the ground water elevation is 16-17 m. The aquifer in this part is belonging to the Nile Alluvium Aquifer. This shallow water table might rise the water table level below Sphinx area (Fig. 5), causing a sever hazards.
- b- Sphinx, Sphinx Temple, Valley Temple of Khafre, Central Field of Mastaba and 159 Khafre Cause Way, this is the most important part of the study where the water appear 160 on the surface at the Valley temple and surrounding area of the Sphinx. The locations of 161 the surveys were chosen according to the limited space approved by the Pyramid 162 163 Archaeological Authority. The locations of the conducted data are shown in (Fig.4). Survey shows a good matching between the different techniques, where the correlation 164 165 between different surveys results, revealed that groundwater elevation between 14-15 m. The base level elevation of the Sphinx Status is 20 m, and safe water table elevation 166 167 should be at elevation of 15 or less. This level is lower than the suburban area of Nazlet El-samman, which might indicate a recharge of the aquifer below Sphinx and increase 168 capillary water rise. 169
- *Sphinx and Sphinx Temple*, GPR-9, SSR-13 and ERI-1 conducted in front of Sphinx and
   Sphinx Temple. The integration of these surveys in front of Sphinx Temple, the
   groundwater elevation is about 14.5-15.5 m, as shown in Figure 6.

*Valley Temple of Khafre and central field of Mastaba*, GPR profiles 3, 4, 5, 10 and 11;
SSR profiles 5, 6, 7, 8 and 14; and ERI 2. The integration of this surveys in front of
Valley Temple of Khafre and central field of Mastaba, the groundwater elevation is about
14-15 m as shown in Figure 7.

- *Tomb of queen Khentkawes*, GPR-11; SSR-15; and ERI-3 conducted near the Tomb.
  Figure 7 shows the surveys conduct near the site. The integration of this surveys in front
  of Valley Tomb of queen Khentkawes, the groundwater elevation is about 14.5-15 m.
- *Valley Temple of Menkaure,* GPR-12; SSR-16; and ERI-4 conducted near the Temple.
  The integration of these surveys in front of Valley Temple of Menkaure, the groundwater
  elevation is about 16.5-17 m. GPR profiles might detect the perched ground water table at
  shallower depth from ground level (Fig. 8).
- *Cause way to Menkaure Pyramid*, show high resistivity value near the surface, and water
   table located at elevation ranges from 22 to 24 m. *Menkaure Queens Pyramids and Menkaure Quarry*, where the surveys in this part conducted at higher topographic relief,
   the correlation of the different techniques revealed that the water table might be
   interpreted at elevations 45-58 m. This might detect the perched ground water table at
   shallower depth from ground level (Figs. 9 and 10).
- 190

191Table 1, shows a comparison of the ground water table elevation data recorded in192some piezometers illustrated in (Fig 12), installed by Cairo University in Wdi Temple193and Sphinx area (AECOM, 2010), and the interpreted water table elevation resulted from194nearest conducted geophysical surveys. There is a relatively good agreement between the195results and differences might be related to the tolerance in the geophysical data and exact196physical properties surface between the wet and saturated media. Moreover, the pumping197stations discharge might lower the water table in the site.

Figure 11 represents a cross-section, using the ERT and GPR data, to illustrate the difference 199 200 of groundwater table elevation between the Great Sphinx to the small pyramids of Menkaure that indicates the increase of groundwater elevation from west to east. As the average water table 201 elevation to be about 15 m, the water table to the west might be considered as perched water 202 table due to leakage, surface runoff and capillary and fracture seepage. Figure 12 represents the 203 204 compiled groundwater table elevation contour map from the geophysical surveys, overlay the groundwater table levels measured from some of the piezometers installed by Cairo University 205 (AECOM 2010). The present geophysical surveys proved that, the pumping system installed by 206

AECOM 2010 lowering the groundwater levels in some piezometer and a need of more pumping
to compensate the recharge of the water leakage resulted from surrounding area of Sphinx.
Figure 13 shows a 3D representation of the groundwater system in Great Giza Pyramids Plateau
and surrounding area.

**211 V. Conclusions**

The integrated interpretation of ERT, SSR and GPR surveys was performed in Great Giza 212 Pyramids site successfully investigate the groundwater aquifer and water table elevation in Great 213 Giza Pyramid and assist the hazards mitigation. An interpreted model consists of three layers 214 assumed to depict the subsurface layers and better delineation of the aquifer layer associated with 215 resistivity range of 40-80  $\Omega$ m and seismic velocity of 1500-1800 m/s. The average water table 216 depth is about 15m, which is safe for the Sphinx status where the base foot at elevation of 20 m. 217 The water table elevation increases in Nazlet Elsamman Village to 16-17m and might recharge 218 the aquifer below Sphinx and Valley Temple which considered a sever hazard on the site. Tomb 219 of Queen Khentkawes threatened by water leakage resulted from vegetation in old cemetery and 220 221 nearby football field. A parched groundwater table might exist in elevated area toward west and southwest. A great care should be taken to the effect of massive urbanization to the west of the 222 Great Giza Pyramids which might affect the groundwater model of the area. The dewatering 223 system should be accomplished to avoid such hazards. 224

**226 Acknowledgements**

Authors would like to thank Prof. Jothiram Vivekanandan, Chief-Executive Editor, Prof. Andrea
Benedetto, the Associate Editor and the reviewer for their constructive comments for improving
our manuscript. Geophysics Department, Cairo University furnished all possible facilities to
conduct the research. IIE-SRF funded the scholarship of S. M Sharafeldin hosted by Geophysical
Engineering Department, KTU, Turkey. Supreme Council of Archaeological authority
permission to conduct the surveys is highly acknowledged.

- 233
- 234
- 235
- 236
- 237
- 238
- 239
- 240
- 241

| 242 References |  |
|-----------------------|--|
|-----------------------|--|

| 243
| Abbas, A. M., Atya, M., EL-Emam, A., Ghazala, H., Shabaan, F., Odah, H., El-Kheder, I., and Lethy, A.: Integrated Geophysical Studies to Image the Remains of Amenemeht- II Pyramid's Complex in Dahshour Necropolis, Giza, Egypt. NRIAG, 2009.
https://www.researchgate.net/publication/234180809. |  |  |  |  |  |  |  |
|---------------------------------|--------------------------------------------------------------------------------------------------------------------------------------------------------------------------------------------------------------------------------------------------------------------------------------------------------|--|--|--|--|--|--|--|
| 248
| Abbas, A. M., El-sayed, E. A., Shaaban, F. A., and Abdel-Hafez, T.: Uncovering the Pyramids Giza Plateau in a Search for Archaeological Relics by Utilizing Ground Penetrating Radar Journal of American Science, 8(2), 1-16, 2012.                                                                    |  |  |  |  |  |  |  |
| 252
| AECOM, ECG, and EDG: Pyramids Plateau Groundwater Lowering Activity. Groundwater Modeling and Alternatives Evaluation. USAID Contract No EDH-I-00-08-00024-00-Order No.02, 2010.                                                                                                                       |  |  |  |  |  |  |  |
| 256
     | Annan, A. P., [2005] Ground-penetrating radar. In Near surface geophysics, Butler DK (ed) Society of exploration geophysicists: Tulsa, Investigations in Geophysics 13, 357-438.                                                                                                                       |  |  |  |  |  |  |  |
| 259
| Azaria, A., Zelt, C. A., and Levander, A.: High-resolution seismic mapping at a groundwate contamination site: 3-D traveltime tomography of refraction data. EGS–AGU–EUG join Assembly, Abstracts from the meeting held in Nice, 2003.                                                                 |  |  |  |  |  |  |  |
| 263
     | Benedetto, A., Tosti, F., Ciampoli, L. B., and D'Amico, F.: An overview of ground-penetrat radar signal processing techniques for road inspections. Signal Processing, 132, 201-209, 2017                                                                                                              |  |  |  |  |  |  |  |
| 266
     | Cassidy, N. J.: Ground penetrating radar data processing, modelling and analysis. In Ground penetrating radar: theory and applications, Jol HM (ed). Elsevier:Amsterdam, 141-176, 2009.                                                                                                                |  |  |  |  |  |  |  |
| 269
| Comas X., Slater L. and Reeve A.: Geophysical evidence for peat basin morphology and stratigraphic controls on vegetation observed in a northern peat land. Journal of Hydrology, 295, 173-184, 2004.                                                                                                  |  |  |  |  |  |  |  |
| 272
| Daniels, D.J.: Ground penetrating radar (2nd edition). The Institution of Electrical Engineers: London, 2004.                                                                                                                                                                                          |  |  |  |  |  |  |  |
| 276
| Dobecki, T. L.: Geophysical Exploration at the Giza Plateau, Egypt a Ten-Year Odyssey.
Environmental & Engineering Geophysical Society (EEGS). 18th EEGS Symposium on the
Application of Geophysics to Engineering and Environmental Problems, 2005.                                             |  |  |  |  |  |  |  |
| 280
| El-Arabi, N., Fekri, A., Zaghloul, E. A., Elbeih, S. F., and laake A.: Assessment of Groundwater Movement at Giza Pyramids Plateau Using GIS Techniques. Journal of Applied Sciences Research, 9(8), 4711-4722, 2013.                                                                                  |  |  |  |  |  |  |  |
| 284
     | Grelle, G. and Guadagno, F. M.: Seismic refraction methodology for groundwater level determination: "Water seismic index". Journal of Applied Geophysics 68, 301–320, 2009.                                                                                                                            |  |  |  |  |  |  |  |

- 287 Griffiths D. H. and King R. F.: Applied geophysics for Engineering and geologists, Pergamon
press, Oxford, New York, Toronto, 221p, 1965.
- 289
Harari, Z.: Ground-penetrating radar (GPR) for imaging stratigraphic features and
groundwater in sand dunes. J. Appl. Geophys., 36, 43–52, 1996.
- 292

Jol, H. M. and Bristow C. S.: GPR in sediments: advice on data collection, basic processing and
interpretation, a good practice guide. In Ground penetrating radar in sediments, Bristow CS and
Jol HM (ed). Geological Society: London, Special Publication 211; 9- 28, 2003.

- Loke, M. H., and Barker, R. D.: Rapid least-squares inversion of apparent resistivity pseudosections by a quasi- Newton method. Geophysical Prospecting, 44 (1), 131–152, 1996.
- 299

- Loke M. H.: Tutorial: 2-D and 3-D electrical imaging surveys. Course Notes, 2012.
- 301
Mahmoud, A. A., and Hamdan, M. A.: On the stratigraphy and lithofacies of the Pleistocene
sediments at Giza pyramidal area, Cairo, Egypt. Sedimentology of Egypt, 10, 145-158, 2002.
  - Neal A.: Ground-penetrating radar and its use in sedimentology: principles, problems and
     progress. Earth science reviews, 66, 261-330, 2004.
  - 307
  - Sandmeier, K. J.: The 2D processing and 2D/3D interpretation software for GPR, reflection
     Seismic and refraction seismic. Karlsruhe, Germany. http://www.sandmeier-geo.de/, 2012.
  - 310
  - Schuster, G. T.: Basics of Exploration Seismology and Tomography. Basics of Traveltime
    Tomography. Stanford Mathematical Geophysics Summer School Lectures. 1998.
    (*http://utam.geophys.utah.edu/stanford/node25.html*).
  - 314
  - Sharafeldin, M., Essa, K.S., Sayıl, N., Youssef, M.S., Diab, Z. E., and Karslı, H.:
    Geophysical Investigation Of Ground Water Hazards In Giza Pyramids And Sphinx Using
    Electrical Resistivity Tomography And Ground Penetrating Radar: A Case Study. Extended
    Abstract, 9th Congress of the Balkan Geophysical Society, Antalya, Turkey. DOI: 10.3997/22144609.201702549, 2017.
  - 320
  - Sheehan, J. R., Doll, W. E., and Mandell, W. A.: An Evaluation of Methods and Available
    Software for Seismic Refraction Tomography Analysis. JEEG, 10 (1), 21–34, 2005.
  - 323
  - Steeples, D. W.: Shallow Seismic Methods. In Y. Rubin, & S. S. Hubbard, Hydrogeophysics (pp: 215-251). Netherlands: Springer, 2005.
  - 326
  - Stipe, T.: A Hydrogeophysical Investigation of Logan, MT Using Electrical Techniques and
    Seismic Refraction Tomography. Degree of Master of Science in Geoscience: Geophysical
    Engineering Option. Montana Tech., 2015.
  - 330

| 331
| Valenta, J., and Dohnal, J.: 3D seismic travel time surveying – a comparison of the time- term method and tomography (an example from an archaeological site). Journal of Applied Geophysics, 63, 46-58, 2007. |  |  |  |  |  |  |  |
|--------------------------|----------------------------------------------------------------------------------------------------------------------------------------------------------------------------------------------------------------|--|--|--|--|--|--|--|
| 335                      | Yehia A.: Geological structures of the Giza pyramids plateau. Middle East Res. Center, Ain                                                                                                                     |  |  |  |  |  |  |  |
| 336                      | Shams Univ., Egypt, Sci. Res. Series, 5, 100-120, 1985.                                                                                                                                                        |  |  |  |  |  |  |  |
| 337                      |                                                                                                                                                                                                                |  |  |  |  |  |  |  |
| 338
| Zelt, A. C., Azaria, A., and Levander, A.: 3D seismic refraction travel time tomography at a groundwater contamination site. Geophysics, 58(9), 1314–1323, 2006.                                               |  |  |  |  |  |  |  |
| 341                      | Zhang, J., and Toksoz, M.: Nonlinear refraction traveltime tomography. Geophysics, 63(5),                                                                                                                      |  |  |  |  |  |  |  |
| 342                      | 1726–1737, 1998.                                                                                                                                                                                               |  |  |  |  |  |  |  |
| 343                      |                                                                                                                                                                                                                |  |  |  |  |  |  |  |
| 344                      |                                                                                                                                                                                                                |  |  |  |  |  |  |  |
| 345                      |                                                                                                                                                                                                                |  |  |  |  |  |  |  |
| 346                      |                                                                                                                                                                                                                |  |  |  |  |  |  |  |
| 347                      |                                                                                                                                                                                                                |  |  |  |  |  |  |  |
| 348                      |                                                                                                                                                                                                                |  |  |  |  |  |  |  |
| 349                      |                                                                                                                                                                                                                |  |  |  |  |  |  |  |
| 350                      |                                                                                                                                                                                                                |  |  |  |  |  |  |  |
| 351                      |                                                                                                                                                                                                                |  |  |  |  |  |  |  |
| 352                      |                                                                                                                                                                                                                |  |  |  |  |  |  |  |
| 353                      |                                                                                                                                                                                                                |  |  |  |  |  |  |  |
| 354                      |                                                                                                                                                                                                                |  |  |  |  |  |  |  |
| 300                      |                                                                                                                                                                                                                |  |  |  |  |  |  |  |
| 357                      |                                                                                                                                                                                                                |  |  |  |  |  |  |  |
| 358                      |                                                                                                                                                                                                                |  |  |  |  |  |  |  |
| 359                      |                                                                                                                                                                                                                |  |  |  |  |  |  |  |
| 360                      |                                                                                                                                                                                                                |  |  |  |  |  |  |  |
| 361                      |                                                                                                                                                                                                                |  |  |  |  |  |  |  |
| 362                      |                                                                                                                                                                                                                |  |  |  |  |  |  |  |
| 363                      |                                                                                                                                                                                                                |  |  |  |  |  |  |  |
| 364                      |                                                                                                                                                                                                                |  |  |  |  |  |  |  |
| 365                      |                                                                                                                                                                                                                |  |  |  |  |  |  |  |
| 366                      |                                                                                                                                                                                                                |  |  |  |  |  |  |  |
| 367                      |                                                                                                                                                                                                                |  |  |  |  |  |  |  |
| 368                      |                                                                                                                                                                                                                |  |  |  |  |  |  |  |
| 369                      |                                                                                                                                                                                                                |  |  |  |  |  |  |  |
| 370                      |                                                                                                                                                                                                                |  |  |  |  |  |  |  |
| 371                      |                                                                                                                                                                                                                |  |  |  |  |  |  |  |
| 372                      |                                                                                                                                                                                                                |  |  |  |  |  |  |  |
| 373                      |                                                                                                                                                                                                                |  |  |  |  |  |  |  |

- Table 1: Average interpreted Groundwater elevations to the nearest 8 piezometers, installed
- 375 piezometers (modified after EACOM 2010)

| Piezom. No.   | Surveyed Area  | Geophysical Data      | Piezom.GWT (m) | Interpreted GWT (m) |
|---------------|----------------|-----------------------|----------------|---------------------|
| PZ-6 & 7      | Nazlet Elsaman | SSR3 & 4 GPR2, 5      | 15.9-17.4      | 16-17               |
| PZ-8          | Sphinx Temple  | SSR3& 4, GPR2 ,5 ERI1 | 15.7           | 14.5-15.5           |
| PZ-11 & 14    | Valley Temple  | SSR14,GPR10 & ERI2    | 14.4 - 14.1    | 14-15               |
| PZ-12, 15 &16 | Sphinx         | SSR13, GPR9 & ERI1    | 15.3-15.6      | 15-15.5             |

Fig. 1: Location map of the study area of Pyramids Plateau.

---

## Referee Comment (RC2) · Anonymous Referee #4 · 15 Nov 2018

The paper deals with an integrated geophysical study to characterize the Giza Pyramids Area. Considering the area in which GPR, ERI and SSR data were collected, the paper could potentially be interesting. Unfortunately, in the present version of the manuscript, data integration and interpretation must be largely improved. The main criticisms concern the interpretation of the geophysical data which is almost totally missing. in what follows, i am listing the main points that have to be adressed:

- The conclusions are not supported by any interpretation of the geophysical results.

- As an example, commenting figure 8, the authors state « GPR profiles might detect the perched ground water table at shallower depth from ground level (Fig. 8) ». Which

are the elements of figure 8 which support this assessment?

- Furthermore, since all the ERI investigations are carried out in the same area, they should be presented by using the same colour range.

- In figures 5-10, the calculated and measured arrival time is shown but I didn't find any reference to it into the text.

- From the SSR results, 3 main strata are clearly present. It is also that their thickness varies area by area. Was this information used to create figures 11 and 13?

- A geological interpretation of the ERTs (ERI) in the text is missing; it is hence not clear how the stratigraphic columns in figure 11 were created.

- Without any interpretation of the geophysical data, the sentence «The present geophysical surveys proved that, the pumping system installed by AECOM 2010 lowering the groundwater levels in some piezometer and a need of more pumping to compensate the recharge of the water leakage resulted from surrounding area of Sphinx. » cannot be supported.

- It is difficult to understand how figure 11, 12 and 13 were created. In figure 13, an inferred fault is also indicated. The authors should show and describe from which kind of data this feature has been inferred.